# Partners in Crime: The Interplay of Proteins and Membranes in Regulated Necrosis

**DOI:** 10.3390/ijms21072412

**Published:** 2020-03-31

**Authors:** Uris Ros, Lohans Pedrera, Ana J. Garcia-Saez

**Affiliations:** Institute for Genetics and Cologne Excellence Cluster on Cellular Stress Responses in Aging-Associated Diseases (CECAD), University of Cologne, Joseph-Stelzmann-Strasse 26, 50931 Cologne, Germany; uris.ros@uni-koeln.de (U.R.); lpedrera@uni-koeln.de (L.P.)

**Keywords:** regulated necrosis, membrane permeabilization, pores, protein-lipid interactions, inflammation

## Abstract

Pyroptosis, necroptosis, and ferroptosis are well-characterized forms of regulated necrosis that have been associated with human diseases. During regulated necrosis, plasma membrane damage facilitates the movement of ions and molecules across the bilayer, which finally leads to cell lysis and release of intracellular content. Therefore, these types of cell death have an inflammatory phenotype. Each type of regulated necrosis is mediated by a defined machinery comprising protein and lipid molecules. Here, we discuss how the interaction and reshaping of these cellular components are essential and distinctive processes during pyroptosis, necroptosis, and ferroptosis. We point out that although the plasma membrane is the common target in regulated necrosis, different mechanisms of permeabilization have emerged depending on the cell death form. Pore formation by gasdermins (GSDMs) is a hallmark of pyroptosis, while mixed lineage kinase domain-like (MLKL) protein facilitates membrane permeabilization in necroptosis, and phospholipid peroxidation leads to membrane damage in ferroptosis. This diverse repertoire of mechanisms leading to membrane permeabilization contributes to define the specific inflammatory and immunological outcome of each type of regulated necrosis. Current efforts are focused on new therapies that target critical protein and lipid molecules on these pathways to fight human pathologies associated with inflammation.

## 1. Introduction

Regulated cell death (RCD) can be triggered under several physiological and pathological conditions; yet, it is fundamental to keep organism homeostasis. Apoptosis is the best-known form of RCD, but in recent years, a number of new genetically encoded machineries leading to cell death have emerged. Pyroptosis, necroptosis, and ferroptosis represent, so far, the best-characterized non-apoptotic forms of regulated cell death [1,2]. Dysregulation of these processes has been linked to many diseases, including autoimmunity, neurodegeneration, and cancer [3,4,5,6]. They also play an important role in the cellular responses to virus and bacterial infections [7]. A common morphological hallmark of these three types of regulated cell death is that they all end up with the rupture of the plasma membrane and release of intracellular contents. Therefore, they have a phenotype similar to accidental necrosis and can cause inflammation at the organism level [6,8,9]. For this reason, these forms of cell death are commonly referred to as regulated necrosis.

In general, the plasma membrane represents the main target for the execution of pyroptosis, necroptosis, and ferroptosis, with the increase in its permeability being a shared key step at the core of regulated necrosis [1,2]. Despite this similarity, each of these pathways is based on a unique machinery leading to specific membrane-related changes. Membrane perforation is facilitated by protein executors in pyroptosis and necroptosis. Specifically, gasdermins (GSDMs) exert pore-forming activity in pyroptosis [10,11], while mixed lineage kinase domain-like (MLKL) protein promotes plasma membrane permeabilization in necroptosis [2,12,13]. In contrast, ferroptosis is special in that membrane damage seems a downstream effect of lipid peroxidation [14]. This diversity of endogenous pro-death effectors ultimately ensures that the nature of the membrane injury will be specific for each type of cell death. Differences in the consequences of the damage will be directly related to its stability, number, and size, as well as the efficacy of counterbalancing membrane repair mechanisms. As the extension of the injury controls the size of molecules that are released through permeabilized membranes, it also has a strong impact on the repertoire of inflammatory and immunological responses that are triggered from cells dying through different mechanisms.

Lipids and membranes have fundamental functions in regulated necrosis. Individual membrane lipids can act as receptors of the protein executors in pyroptosis and necroptosis or as modulators of the properties of the plasma membrane required for their activity [15,16,17]. The unique properties of the plasma membrane provide a suitable environment for conformational reorganization and oligomerization of GSDMs and MLKL, two pre-requisites for their permeabilizing activity. On the other hand, alteration of lipid metabolism and remodeling of membrane lipids by peroxidation are the most distinctive attributes of ferroptosis [18,19]. Besides the increase of plasma membrane permeability, other alterations such as lipid remodeling, exosome release and membrane repair are common features of pyroptosis, necroptosis, and ferroptosis [2]. Among them, membrane repair has a central role in restricting the extension of membrane damage and, therefore, modulating the processes activated downstream of or in parallel to membrane permeabilization [20,21,22]. In this chapter, we focus on the essential interplay existing between membranes and the protein and lipid executors of pyroptosis, necroptosis, and ferroptosis. We highlight the mutual relevance of protein–protein and protein–lipid interactions during regulated necrosis.

## 2. Plasma Membrane Permeabilization: A Common Feature of Regulated Necrosis

### 2.1. Pore Formation by GSDMs in Pyroptosis 

Pyroptosis is a highly inflammatory type of regulated cell death that is associated with the formation of membrane pores by GSDMs and often involves activation by inflammatory caspases 1 or 11 within the inflammasome complex [1] (Figure 1). Two different inflammasome pathways, referred to as canonical or non-canonical, can lead to pyroptosis. In the canonical one, danger signals (e.g., pathogen- or damage-associated molecular patters (PAMPs and DAMPs)) are sensed by the inflammasome through pattern recognition receptors. After its activation within the inflammasome, caspase 1 mediates the cleavage and activation of GSDMD and of the inflammatory cytokines IL-1β and IL-18. In the non-canonical pathway, intracellular lipopolysaccharide (LPS) binds to caspase 4 and 5 (in human) and 11 (in mouse), resulting in their oligomerization and activation. These inflammatory caspases cleave a conserved site of GSDMD. In both the canonical and non-canonical pathways, the caspase-mediated proteolytic cleavage of GSDMD induces the dissociation of the N-terminal domain (GSDMD-N) from its auto-inhibitory C-domain. GSDMD-N has the ability to translocate to the inner leaflet of the plasma membrane, where it oligomerizes and induces the opening of membrane pores [1,6]. Besides GSDMD, other members of the GSDM family including GSDMA, GSDMB, GSDMC, and GSDME display pore-forming and pyroptotic activity upon the release of their N-terminal domain, although their mechanism of activation and the cellular settings under which they act are less clear [6].

Specific membrane lipids are needed for GSDMD engagement at the plasma membrane. In particular, GSDMD-N has a high affinity for acidic phospholipids such as phosphatidylinositol phosphates (PIPs) and cardiolipin (CL), and it can also bind to phosphatidic acid (PA) and phosphatidylserine (PS) [10,23]. A similar lipid-binding pattern has been identified for other members of the GSDMs family [10,23,24]. Since PIPs, PA, and PS are mostly present in the inner leaflet of the plasma membrane, the N-terminal domain of GSDMs can only cause death from inside the cells. Some experimental data suggests that GSDMD-N can also disrupt mitochondria and inhibit bacterial growth. This activity is probably associated with the presence of CL in the inner membrane of mitochondria in eukaryotes and in bacterial membranes [10,23]. Other lipids such as sphingomyelin and cholesterol also impact on the activity of GSDMD-N. While sphingomyelin promotes binding of GSDMD-N to lipid vesicles, cholesterol has the opposite effect. These two lipids are well-known modulators of membrane fluidity and the formation of membrane boundaries or microdomains [25]. Therefore, they probably act by regulating the insertion of GSDMD-N into the bilayer by changing the physical properties of the membranes [23,25,26].

Given the high sequence homology of the N-terminal domain of GSDMs, it is very likely that these proteins follow a similar mechanism for pore formation [6]. The lipid membrane provides a suitable environment to trigger the conformational changes required for their structural evolution from soluble monomers to membrane-embedded oligomers [27]. In recent years, our understanding of the mechanism of GSDMs pore formation has dramatically increased with the structural characterization of different variants in solution [10] and in membranes [27,28]. The analysis of the crystal structure of full-length GSDMA3 [10] and the cryo-electron microscopy structure of a GSDMA3-N pore isolated from reconstituted liposomes [27] shed light on the mechanisms of membrane insertion and pore assembly. From this data, it emerged that in solution, the N-terminal domain is kept inhibited due to inter-domain interactions with the C-terminal domain. Auto-inhibition can be abrogated by cleavage by inflammatory caspases or, potentially, by any other molecular events hindering this inter-domain interaction [6]. Free monomers of GSDM-N can bind to the membrane through negatively charged lipids, which drives the exposure of new oligomerization interfaces and the assembly of a β-barrel pore. Pore formation is a dynamic process involving heterogeneous structures. High-resolution atomic force microscopy studies with GSDMD-N showed that monomers can first associate forming arcs or slit structures that later evolve into transmembrane pores with a ring shape. Notably, all these structures are able to perforate the membranes forming holes in the 10–30 nm range [28]. The fact that incomplete rings of GSDMD-N promote the opening of the membrane clearly indicates the direct involvement of lipids in pore opening. Indeed, the lipid contribution to toroidal-like pore structures and shape heterogeneity are two properties reminiscent of the pores formed by the apoptotic executor Bax and the membrane attack complex perforin-like/cholesterol-dependent cytolysin (MACPF/CDC) family [29,30].

GSDMs pores are non-selective membrane channels characterized by a relatively large size. Therefore, they can allow the passage of different ions such as K^+^ and Ca^2+^, interleukins, and other small cytosolic proteins, even before cell lysis [31,32]. In fact, movement of molecules through these pores under sub-lytic conditions has been proposed to impact on different intracellular signaling pathways and/or the immunological outcome of pyroptotic cells [6]. Notably, K^+^ and Ca^2+^ are mediators for NLRP3 inflammasome activation, contributing not only to the osmotic imbalance driving to cell lysis but also to the activation of the pyroptotic pathway in a positive feedback/amplification mechanism [2,33,34]. In response to an increase of cytosolic Ca^2+^, the endosomal sorting complex required for transport (ESCRT) machinery is activated and protects from GSDMD-N pores at the plasma membrane [22]. Most probably, this membrane repair mechanism has an active role in regulating the number and size of pores at the membrane, which ultimately defines the cell fate. If the pores are repaired, the cells could escape from death and the release of specific inflammatory cytokines such as IL-1β and IL-18, and DAMPs would be transient and highly controlled. In the opposite scenario, the increased number of GSDMD-N pores at the membrane would lead to osmotic imbalance and/or the loss of function of intracellular organelles and consequently plasma membrane bursting and cell death. That said, it is possible that cell lysis might not always be the main function of GSDMs pores, as they have been proposed to be capable of releasing mature cytokines in a cell death-independent manner [6]. Future studies should clarify the conditions determining cell death-independent and -dependent effects of GSDMs pore formation and their impact on the strength of the inflammatory responses derived from pyroptotic cells. 

### 2.2. Membrane Permeabilization by MLKL in Necroptosis

Necroptosis is a caspase-independent form of regulated cell death that critically depends on the pseudokinase MLKL (Figure 2) [1]. One molecular requirement for necroptosis is the suppression of caspase 8 activity, which hinders the activation of the extrinsic apoptotic pathway [1,4]. The protein machinery mediating necroptosis can be triggered through the activation of death receptors (e.g., tumor necrosis factor receptor 1 (TNFR1)) or pathogen recognition receptors (e.g., Toll-like receptor 3/4 (TLR3/4)). These pathways converge in the formation of the necrosome, a cytosolic protein complex containing the receptor interacting protein kinase 3 (RIP3). RIP3 homo-oligomerization within the necrosome results in its activation by auto-phosphorylation and recruitment of MLKL. RIP3-mediated phosphorylation then drives MLKL oligomerization and translocation to the plasma membrane, where it executes its necroptotic activity leading to plasma membrane permeabilization. MLKL is the main player in necroptosis and its most downstream executor discovered so far [12,35,36]. 

Lipids and membranes are essential pieces of the necroptotic machinery that reshape MLKL from the cytosolic inactive state to the membrane-bound active form. MLKL inserts into the plasma membrane in a multi-step process [37,38]. Upon activation, MLKL undergoes a conformational change that first facilitates its weak interaction with PIPs. Then, as a result of the contact to the membrane environment, new high-affinity sites are exposed and stronger interactions with the lipid bilayer are stabilized [37]. In addition, inositol phosphates (IPs), which are soluble intermediates of lipid metabolism, also modulate MLKL necroptotic activity (Figure 2). Specifically, the highly phosphorylated versions (i.e., IP6 > IP5 > IP4) can bind to MLKL, thereby promoting a conformational change that exposes the killer (4HB) domain by releasing the auto-inhibition by the adjacent brace regions [39,40]. In addition, it has been found that very-long-chain fatty acids promote necroptosis, probably due to their effect on membrane organization that would potentially modulate MLKL activity [15]. On the other hand, lipid rafts have been proposed to act as recruiting sites for the removal of phosphorylated MLKL from the membrane during membrane repair [41,42,43]. 

Despite the essential role of MLKL in necroptosis, the mechanism how it executes membrane permeabilization remains a puzzle. Competing models propose that plasma membrane permeabilization could be either indirectly or directly mediated by MLKL (Figure 2). Early studies proposed that MLKL could activate either Ca^2+^ or Na^+^ endogenous channels [41,44], ion imbalance being ultimately responsible for cell death. However, genetic deletion of specific ion channels had a minor effect on protecting cells against necroptosis [44,45]. Most likely, MLKL oligomers directly participate in the alteration of ion fluxes across the plasma membrane in necroptosis, either as a result of its partial insertion into the lipid bilayer [46] or acting as selective channels or pores [47]. However, so far, all experimental evidence regarding the ability of MLKL to form membrane pores has been obtained using artificial lipid vesicles [38,48]. These experiments required protein concentrations that are physiologically too high, which questions the potential pore-forming activity of MLKL. Moreover, the use of recombinant versions of MLKL that do not fully mimic its active state in cells has also limited the implications of these in vitro experiments. In a cell-based study, we found that necroptosis is mediated by the formation of nanopores in membranes. Even though MLKL has not been yet directly associated with these structures, our findings positioned pore formation as a core mechanism in necroptosis [49]. 

MLKL action is linked to the enhancement of ion fluxes (e.g., Ca^2+^, Mg^2+^, Na^+^, and K^+^) across the plasma membrane [41,44,47]. Chief among them, Ca^2+^ is of special interest due to its central role in cell death and survival. There is mounting evidence showing that cytosolic Ca^2+^ concentration increases upon necroptosis induction, a process that depends on MLKL and takes place much earlier than plasma membrane bursting and cell death [2,13,49]. During this time, cytosolic Ca^2+^ could activate several intracellular processes that could lead to cell death or survival. Related to MLKL activity at the plasma membrane and the increase in cytosolic Ca^2+^, necroptotic cells can expose phosphatidylserine (PS) on its outer leaflet [2], which mediates their recognition by macrophages [13,20].

Necroptotic cells in which MLKL appears phosphorylated at the membrane are not necessarily committed to death. During this early phase of necroptosis, a number of cellular alterations take place, which are believed to play a role in the communication of dying cells with the surroundings in a specific manner. Exosomes release, membrane repair, targeting of intracellular organelles, and production of inflammatory cytokines and DAMPs are some of the processes that are activated during the commitment phase of necroptosis [13,50,51]. As described before in pyroptosis, increase of cytosolic Ca^2+^ plays a key role in the activation of the ESCRT-mediated membrane repair mechanism that removes phosphorylated MLKL and damaged sections from the plasma membrane [20]. Other membrane repair machineries such as flotillin-mediated endocytosis followed by lysosomal degradation and ALIX-syntenin-1-mediated exocytosis also counterbalance necroptosis [42]. Whether the result of MLKL activation is that the cell dies or survives probably depends on the extension of MLKL-dependent membrane damage and the efficiency of the opposite machineries that counteract its assembly at the plasma membrane. In line with these findings, an interesting possibility would be that MLKL-mediated membrane reshaping is not necessarily meant to kill cells but to promote a number of cellular alterations that prepare necroptotic cells to respond in a unique manner.

### 2.3. Lipid Peroxidation in Membranes during Ferroptosis

Ferroptosis is a caspase-independent form of regulated necrosis that involves the generation of lipid peroxides through iron-dependent reactions (Figure 3) [14,52]. In contrast to pyroptosis and necroptosis, which possess a specific protein machinery that mediates plasma membrane disruption, no specific protein has been identified so far involved in cell death execution in ferroptosis. Instead, ferroptosis is induced when lipid peroxidation overwhelms the cellular antioxidant defense. Loss of activity of glutathione peroxidase 4 (GPX4) results in the accumulation of lipid peroxides in membranes, which leads to the disruption of plasma membrane integrity and cell death [14,19,53,54]. GPX4 is a unique enzyme that plays a central role in antioxidant mechanisms of the cell, since it mediates the reduction of toxic peroxidized phospholipids in membranes to less toxic lipid alcohols [19]. In addition to the canonical glutathione-based GPX4 pathway, the ferroptosis suppressor protein 1 (FSP1) targets ubiquinone (CoQ_10_) in the plasma membrane and reduces it to ubiquinol, which protects cells from lipid peroxidation due to its capacity of trapping lipid radicals [55,56]. The GTP cyclohydrolase-1 (GCH1) and its metabolic derivative tetrahydrobiopterin/dihydrobiopterin also inhibit ferroptosis by modulating CoQ_10_ availability. Additionally, GCH1 plays this role by selectively preventing depletion of phospholipids with two polyunsaturated fatty acids PUFAs [57].

Specifically, the metabolism of PUFAs and the collapse of cellular redox homeostasis have been identified as key processes in ferroptosis [52,58,59]. PUFAs are more sensitive to oxidation, since they contain labile bis-allylic hydrogens. Membrane phospholipids containing long-chain PUFAs such as arachidonic (20:4), adrenic acid (22:4), and docohexaenoic acid (22:6) are specifically peroxidized during ferroptosis [14,19,60,61]. Consistent with this, addition of PUFAs to different cells sensitizes them to ferroptosis due to their increased levels in membranes [57,59]. In contrast, exogenous monounsaturated fatty acids suppress ferroptosis by replacing PUFAs from the cellular membranes [62]. Among the different membrane phospholipids, specific oxidation of phosphatidylethanolamine-containing PUFAs has been identified as a product of ferroptosis [60,61]. Specific lysophospholipids, such as lysophosphatidylcholine, lysophosphatidylethanolamine, and lysophosphatidylinositol, also accumulate in membranes of ferroptostic cells [57,63]. This accumulation probably results from the phospholipase A2-mediated cleavage of peroxidized PUFAs from the glycerophospholipid backbones [9,19]. Furthermore, triacylglycerols and diacylglycerols containing PUFA tails are depleted in cells undergoing ferroptosis [57,63,64,65]. 

Classic enzymes of lipid metabolism have been linked to ferroptosis through their essential role in the modification of membrane phospholipids. Acyl-CoA synthetase long-chain family 4 (ACSL4) and lysophosphatidylcholine acyltransferase 3 (LPCAT3) facilitate ferroptosis by mediating the incorporation of Acyl-CoA-PUFAs into membrane phospholipids (Figure 3) [9,60,61]. Genetic deletion of these genes fully prevented ferroptosis in tamoxifen-induced GPX4-knock out cells [61]. In addition, lipoxygenases ALOX12 and -15 have been associated to ferroptosis due to their function as iron-dependent dioxygenases that catalyze the incorporation of oxygen into the long-chain PUFAs [59,66]. However, a recent study proposed that peroxidized lipids are mainly derived from non-enzymatic reactions and that lipoxygenases just have a secondary role in ferroptosis [9]. 

Ferroptosis intrinsically involves the alteration of the properties of membrane phospholipids. However, there is scarce information about the membrane changes that are induced in this form of cell death. In addition, the specific membrane target of lipid peroxidation during ferroptosis is a matter of debate. Both plasma and mitochondrial membranes have been identified as potential main sites of lipid peroxidation in ferroptotic cells. Mitochondrial fragmentation, cristae disassembly, and mitochondrial outer membrane permeabilization are some of the dramatic changes in mitochondria morphology observed during ferroptosis [14,61]. Recently, we proposed that plasma membrane damage in ferroptosis involves the formation of pores of few nanometers in size that are linked to the increase of cytosolic Ca^2+^ [67]. In agreement with its role in other types of regulated necrosis, increase in cytosolic Ca^2+^ upon membrane damage correlated with the activation of ESCRT-mediated membrane repair, which had a pro-survival role by counterbalancing the kinetics of cell death and modulated the inflammatory signature of ferroptosis. Despite these findings, there is still no evidence about the protein- or lipid-based nature of these pores. It is a matter of strong debate whether peroxidation of PUFAs at the plasma membrane would be sufficient to cause its permeabilization. It has been hypothesized that peroxidized phospholipids and their derivate lysophospholipids could be directly toxic to cells through the formation of small discontinuities in the plasma membrane involving membrane thinning and increase of membrane curvature [58] (Figure 3). Alternatively, lipid peroxidation could modulate the activity of membrane proteins, indirectly triggering membrane disruption. All these are interesting hypotheses that should be experimentally validated in the future.

## 3. Conclusions and Perspectives

Our understanding of pyroptosis, necroptosis, and ferroptosis as forms of regulated necrosis has improved largely in the last years. They all culminate with dismantling of the plasma membrane and therefore have in common the ability to activate inflammation and adaptive immune responses. However, the molecular details how these forms of cell death are executed and the precise mechanisms leading to the increase in plasma membrane permeability are specific for each type of cell death. While membrane damage is mediated by protein effectors in pyroptosis and necroptosis, lipid peroxidation leads to membrane disruption in ferroptosis. These different molecular executors determine the type of injury that is infringed to the membrane and the downstream consequences of membrane damage. Non-selective and relatively large GSDMs pores in pyroptosis allow the release of intracellular molecules such as pro-inflammatory interleukins, which act as activators of the immune system [10]. Evidence suggests that MLKL forms small and selective pores or channels that mediate the movement of ion fluxes but not big molecules in necroptosis [47,49]. In ferroptosis, the exact mechanism of membrane permeabilization is less clear but also seems to involve pore formation in the plasma membrane [67]. Therefore, pore formation has emerged as a common feature underlying different types of regulated necrosis. As an opposing mechanism, membrane repair by the ESCRT machinery counterbalances all these types of cell death by removing the damage sections of the plasma membrane [20,22,67].

The plasma membrane is the final target of regulated necrosis and at the same time the 2D scaffold where protein executors can assemble to perform their function. In this regard, membrane lipid composition and its biophysical properties might strongly determine specific protein–protein and protein–lipid interactions required for the activation of the death effectors. In pyroptosis and necroptosis, membrane binding of GSDMD and MLKL takes places through protein interaction with specific anionic phospholipids. Membrane interactions are essential for the reshaping of proteins and the exposure of new interaction sites, leading to the assembly of functional oligomers, which finally punch the membrane. A special case is ferroptosis, which is dictated by alterations of lipid metabolism driving to the accumulation of lipid peroxides in cellular membranes. In this case, the obvious modification of the properties of the plasma membrane seems to affect both its integrity and the function of protein components of the bilayer.

Altogether, it is clear that protein–protein and protein–lipid interactions, as well as plasma membrane dynamics, play essential roles in the regulation and execution of pyroptosis, necroptosis, and ferroptosis. As these forms of RCD are implicated in diverse pathologies, efforts are focused on the understanding of the molecular events at the plasma membrane that are relevant for these types of cell death. Improving our understanding of these processes holds potential to assist the development of new therapies that target membranes and lipid metabolic processes to fight human pathologies linked to regulated necrosis.

## Figures and Tables

**Figure 1 ijms-21-02412-f001:**
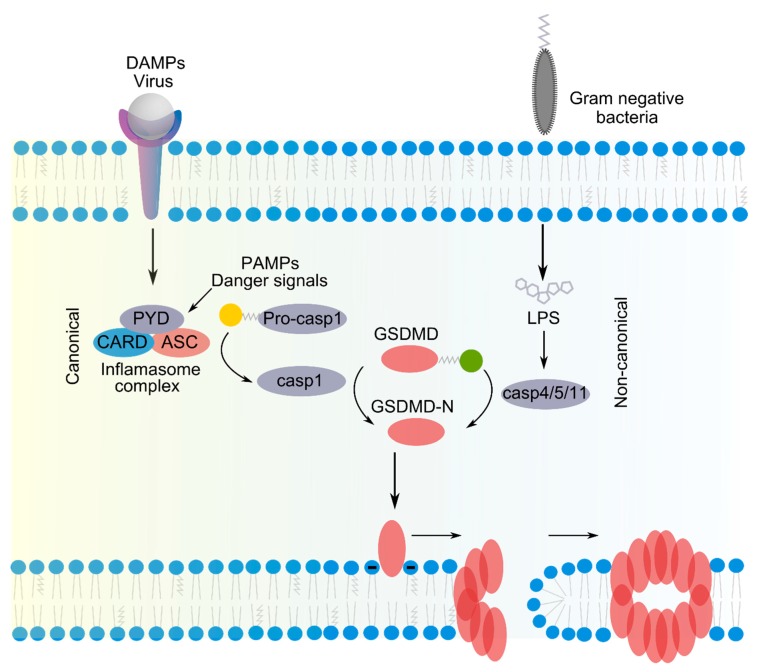
Pyroptosis is a caspase-dependent form of regulated necrosis that critically depends on the pore forming activity of the gasdermins (GSDMs). Two different pathways can trigger pyroptosis. In the canonical one, external or endogenous danger signals are sensed by the inflammasome through pattern recognition receptors that interact with procaspase-1 either directly through a PYRIN-PAAD-DAPIN (PYD)/C-terminal caspase-recruitment domain (CARD) (PYD/CARD) domain or indirectly via an Apoptosis-associated speck-like protein containing a CARD (ASC) adaptor. The yellow ball represents the protein domain removed during the activation of caspase 1. Active caspase 1 cleaves and activates the inflammatory cytokines IL-1β and IL-18. In the non-canonical pathway, lipopolysaccharide (LPS) from Gram-negative bacteria binds to the CARD domain of caspase 4, 5, and 11. In both pathways, caspases mediate the cleavage of GSDMD, which allows the release of the N-terminal domain (GSDMD-N) from auto-inhibition by the C-domain (represented by a green ball). GSDMD-N translocates to the inner leaflet of the plasma membrane, where it interacts with anionic phospholipidsand oligomerizes forming arc or slits structures that evolve to a ring of a β-barrel pore. Blue balls are the polar head groups of membrane phospholipids.

**Figure 2 ijms-21-02412-f002:**
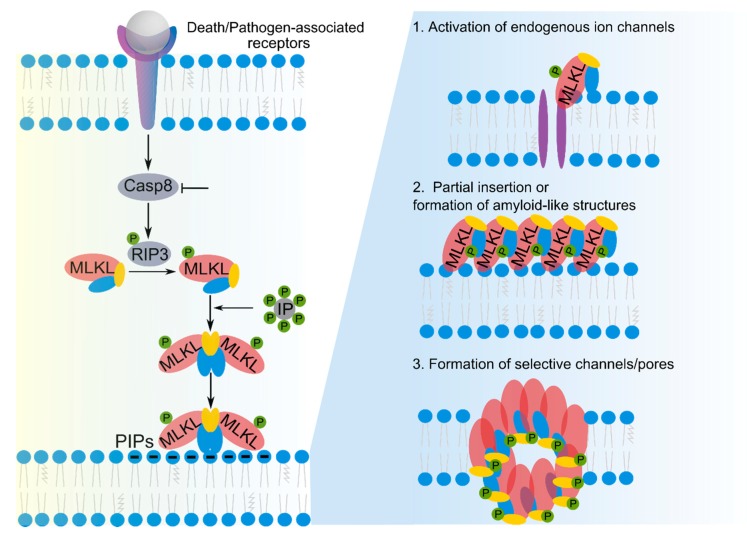
Necroptosis is a caspase-independent form of regulated necrosis that critically depends on membrane permeabilization of mixed lineage kinase domain-like (MLKL) protein. Left: Necroptosis is initiated by various stimuli that all result in the formation of the necrosome, phosphorylation of MLKL, and its subsequent translocation to the plasma membrane. MLKL oligomerization is also modulated by its interaction with highly phosphorylated inositol phosphates (IPs), which are soluble products of lipid metabolism. MLKL activation facilitates its weak interaction with phosphatidylinositol phosphates (PIPs) at the membrane. Membrane binding promotes the exposure of new high-affinity sites and strong membrane interaction. Right: Different models of MLKL-mediated membrane permeabilization in necroptosis: (1) indirect activation of endogenous ion channels, (2) partial insertion in the bilayer forming amyloid fibers, and (3) formation of selective channels or pores.

**Figure 3 ijms-21-02412-f003:**
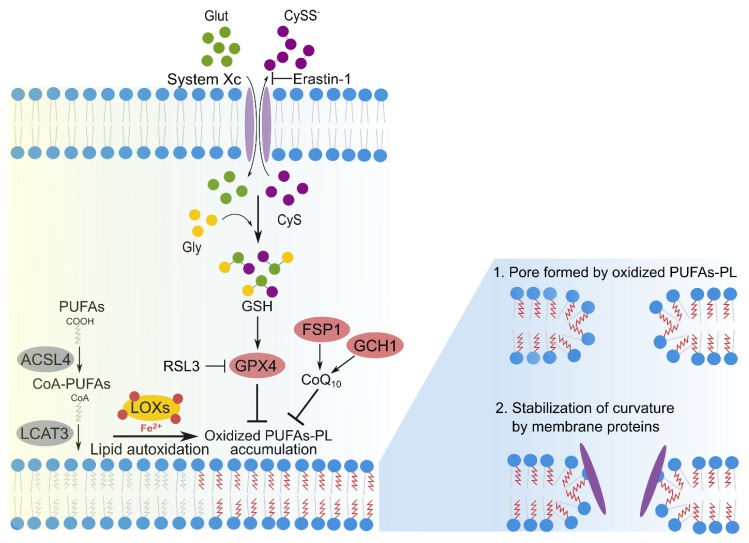
Ferroptosis is a caspase-independent form of regulated necrosis that involves the production of phospholipid peroxides in cellular membranes. Left: Loss of glutathione peroxidase 4 (GPX4) activity is the main trigger of ferroptosis. Ras-selective Lethal Small Molecule 3 (RSL3), which directly inhibits GPX4, and erastin-1, which affects the catalytic cycle of GPX4 by perturbing the levels of glutathione (GSH), both trigger ferroptosis. Ferroptosis suppressor protein 1 (FSP1) and GTP cyclohydrolase-1 (GCH1) that target ubiquinone in the plasma membrane can regulate accumulation of membrane oxidized PUFAs-phospholipids. Acyl-CoA synthetase long-chain family 4 (ACSL4) and lysophosphatidylcholine acyltransferase 3 (LPCAT3) mediate the incorporation of PUFAs into membrane. Lipoxigenases (LOX) catalyzes the incorporation of oxygen into the PUFAs, to produce lipid hydroperoxides, which are accumulated at the membrane in the absence of GPX4 activity. Right: Different models of plasma membrane permeabilization in ferroptosis: induction of membrane curvature by the accumulation of phospholipid hydroperoxides (1) (2) alteration of the function of membrane proteins and/or stabilization of membrane curvature by endogenous proteins.

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
