# Peer review of "Partners in Crime: The Interplay of Proteins and Membranes in Regulated Necrosis"

_ijms, 2020, doi:10.3390/ijms21072412_

Round 1

Reviewer 1 Report

The manuscript showed the interaction between protein and membrane components are essential processes during regulated necrosis. However, there are few major issues need to be considered and required correction:

Page 1: This abstract statement said more about the objectives however, the abstract lacks the finding, conclusion along with the future direction.

Page 2: A major section missing before starting to talk about an overview of Plasma membrane permeabilization, that is how this critical manuscript has been constructed? The manuscript should include the selection criteria such as the use of what kind of data bases sources used (Such as Scopus, MEDLINE, PubMed, Cochrane and ScienceDirect etc.), how did the filter occur (how many years of studies, and types study).

The manuscript lacks the table which may show the comparative analysis of the past studies, while figures were descriptive and showed hypothetical mechanism.

Author Response

We have read and discussed the comments from the reviewers and thank them for their time and suggestions. Please find below the point-by-point to the comments from reviewer 1:

Page 1: This abstract statement said more about the objectives however, the abstract lacks the finding, conclusion along with the future direction.

We have changed the abstract as suggested. It is now as follows:

Pyroptosis, necroptosis and ferroptosis are well-characterized forms of regulated necrosis that have been associated with human diseases. During regulated necrosis, plasma membrane damage facilitates the movement of ions and molecules across the bilayer, which finally leads to cell lysis and release of intracellular content. Therefore, these types of cell death have an inflammatory phenotype. Each type of regulated necrosis is mediated by a defined machinery comprising protein and lipid molecules. Here, we discuss how the interaction and reshaping of these cellular components are essential and distinctive processes during pyroptosis, necroptosis and ferroptosis. We point out that, although the plasma membrane is the common target in regulated necrosis, different mechanisms of permeabilization have emerged depending on the cell death form. Pore formation by Gasdermins (GSDMs) is a hallmark of pyroptosis, while Mixed Lineage Kinase domain-Like (MLKL) facilitates membrane permeabilization in necroptosis and phospholipid peroxidation leads to membrane damage in ferroptosis. This diverse repertoire of mechanisms leading to membrane permeabilization contributes to define the specific inflammatory and immunological outcome of each type of regulated necrosis. Current efforts are focused on new therapies that target critical protein and lipid molecules on these pathways to fight human pathologies associated to inflammation.

Page 2: A major section missing before starting to talk about an overview of Plasma membrane permeabilization, that is how this critical manuscript has been constructed? The manuscript should include the selection criteria such as the use of what kind of data bases sources used (Such as Scopus, MEDLINE, PubMed, Cochrane and ScienceDirect etc.), how did the filter occur (how many years of studies, and types study).

The manuscript lacks the table which may show the comparative analysis of the past studies, while figures were descriptive and showed hypothetical mechanism.

The aim of this review paper is to summarize the state of art in the topic of regulated necrosis with an emphasis in the interplay of proteins, lipids and membranes, which is a fundamental aspect in the field of cell death and inflammation. For this, we searched into the available literature which is summarized in the reference list, and provide an authoritative overview of the topic. In our review, we carry out an extensive and novel analysis of the current published findings and waved the available information with a sense of judgment, pointing out our own opinion about the latest developments and newest insights in the field. Due to the common interest of the wide community of life scientists on the topic of regulated necrosis, we wrote the review intending to be useful for a broad audience, following the common format for review papers in life sciences. Therefore, we would like to keep the current format of our review.

Reviewer 2 Report

The authors in work entitled :” Partners in crime: the interplay of proteins and membranes in regulated necrosis” reviewed the importance of membrane, lipids and protein interactions in pyroptosis, necroptosis and ferroptosis as the forms of programmed cell death. The information are new and relevant in the domain.

The review is well discussed and supported with recent literature. I have found only few formatting issues. E.g. ref. 68 is not complete.

Author Response

We thank the reviewer for his/her positive comments. We corrected the reference 68 as suggested.

Round 2

Reviewer 1 Report

I totally agree with process and content of summarization however it should be noted down as method which I have suggested in the previous comment. Still lack those parameters in the manuscript.